# Effects of Cadmium Stress on Tartary Buckwheat Seedlings

**DOI:** 10.3390/plants13121650

**Published:** 2024-06-14

**Authors:** Hanmei Du, Lu Tan, Shengchun Li, Qinghai Wang, Zhou Xu, Peter R. Ryan, Dandan Wu, An’hu Wang

**Affiliations:** 1Panxi Featured Crops Research and Utilization Key Laboratory of Sichuan Province, Xichang University, Xichang 615000, China; tanlu19910222@163.com (L.T.); 18995853890@163.com (S.L.); wqh1226823@163.com (Q.W.); xzhbiol@163.com (Z.X.); 2Division of Plant Sciences, Research School of Biology, The Australian National University, Canberra, ACT 2601, Australia; peterryan1959@outlook.com; 3Triticeae Research Institute, Sichuan Agricultural University, Chengdu 611130, China; 14646@sicau.edu.cn

**Keywords:** Tartary buckwheat, cadmium toxicity, growth, cell structure, oxidative stress

## Abstract

Cadmium (Cd) is a naturally occurring toxic heavy metal that adversely affects plant germination, growth, and development. While the effects of Cd have been described on many crop species including rice, maize, wheat and barley, few studies are available on cadmium’s effect on Tartary buckwheat which is a traditional grain in China. We examined nine genotypes and found that 30 µM of Cd reduced the root length in seedlings by between 4 and 44% and decreased the total biomass by 7 to 31%, compared with Cd-free controls. We identified a significant genotypic variation in sensitivity to Cd stress. Cd treatment decreased the total root length and the emergence and growth of lateral roots, and these changes were significantly greater in the Cd-sensitive genotypes than in tolerant genotypes. Cd resulted in greater wilting and discoloration in sensitive genotypes than in tolerant genotypes and caused more damage to the structure of root and leaf cells. Cd accumulated in the roots and shoots, but the concentrations in the sensitive genotypes were significantly greater than in the more tolerant genotypes. Cd treatment affected nutrient uptake, and the changes in the sensitive genotypes were greater than those in the tolerant genotypes, which could maintain their concentrations closer to the control levels. The induction of SOD, POD, and CAT activities in the roots and shoots was significantly greater in the tolerant genotypes than in the sensitive genotypes. We demonstrated that Cd stress reduced root and shoot growth, decreased plant biomass, disrupted nutrient uptake, altered cell structure, and managed Cd-induced oxidative stress differently in the sensitive and tolerant genotypes of Tartary buckwheat.

## 1. Introduction

Industrialization has increased the deposition of many harmful pollutants in the environment. The heavy metal cadmium (Cd) became a serious environmental concern as a result of many industrial processes as well as a contaminant in mine tailings, wastewater irrigation, atmospheric deposition, and sewage sludge [1,2]. According to a survey of soil pollution in China, 16.1% of arable land had been polluted, and Cd accounts for the highest proportion of all pollutants [3]. Compared with other metal contaminants, Cd is easily absorbed, transported, and accumulated in plant organs. Therefore, even low concentrations of Cd are potentially toxic to plants and threaten agricultural production [4,5,6]. Moreover, Cd is a known carcinogen, and its accumulation in edible plants poses a risk to human health by inducing cancers and affecting bone development and kidney function [7,8]. 

Cd accumulates in roots, shoots, flowers, fruits, and other organs and adversely affects many aspects of development and metabolism from germination through to maturity [1,9,10]. Cd accumulation in several crops, including rice, wheat, and tomato decreases root and shoot growth and inhibits the absorption of water and nutrients [11,12,13,14,15]. The inhibition of root growth is one of the early symptoms of Cd stress; the effects on cell division and elongation alter root morphology, perhaps by disrupting auxin and cytokinin signaling [16,17,18]. Cd stress triggers oxidative stress, damaging membranes and altering organelle development [19,20,21]. The Cd-induced accumulation of reactive oxygen species such as superoxide anion (O_2_^−^) and hydrogen peroxide (H_2_O_2_) has been measured in *Arabidopsis*, *Brassica juncea*, rice, and apple [22,23,24].

Certain species, and even genotypes within species, induce defensive mechanisms to protect them from Cd stress. These can be broadly divided into avoidance mechanisms and tolerance mechanisms [25]. Avoidance mechanisms reduce Cd entry into the cytosol to maintain lower tissue concentrations by reducing transport processes or binding Cd in the cell wall. The tolerance mechanism helps to moderate the stress once the metal ions have entered the cytosol better by chelating the heavy metals in less harmful complexes, compartmentalizing the ions to less sensitive organelles such as the vacuole, or by activating antioxidative defense pathways. Antioxidant enzymes are involved with mitigating oxidative damage caused by a wide range of abiotic stresses apart from heavy metals including drought and mineral toxicities [26].

Tartary buckwheat (*Fagopyrum tataricum* (L.) Gaertn) is an important rain-fed crop in China, Japan, Korea, and India. Its valuable contribution to diet and traditional medicines [27] is likely related to its high content of flavonoids, amino acids, dietary fiber, and other nutrients [28,29]. While previous studies show that the cultivation, yield, and quality of Tartary buckwheat are affected by Cd stress [30,31], few studies have focused on the earlier effects of Cd on Tartary buckwheat seedlings. Furthermore, the mechanism(s) by which some cultivars of Tartary buckwheat can resist Cd stress better than others remains unclear.

This study investigated the effects of Cd on a range of Tartary buckwheat cultivars. We aimed to reveal the physiological effects of Cd stress on early growth and examine the molecular mechanisms that could be contributing to the greater tolerance to Cd exhibited by some genotypes.

## 2. Results

### 2.1. Cd Stress Inhibits the Growth of Tartary Buckwheat Seedlings

Treatments with 30 μM of CdCl_2_ inhibited the shoot and root growth in all nine *F. tataricum* genotypes compared with controls, but some genotypes were significantly more sensitive than others. The inhibition of root length (RL) among the nine genotypes varied between 4 and 30%; the reduction in shoot dry weight (SDW) varied between 6 and 30%, and the reduction in root dry weight (RDW) varied between 4 and 42% (Figure 1B–D). The growth of the B6, B7, B8, and B9 genotypes was inhibited by 30 μM of CdCl_2_ to a greater extent than the growth of B1, B2, and B3, while the effects on B4 and B5 were more moderate. Among these, B6 had the greatest inhibition in RL and RDW, while B7 showed the greatest reduction in SDW. The total dry weights (DWs) of B6, B7 B8, and B9 after Cd treatment were 70%, 69%, 71%, and 74% of that of controls, respectively, whereas DW of those of B1, B2, and B3 were 90%, 92%, and 93% of that of the controls, respectively. These results indicate that B1, B2, and B3 displayed greater resistance to Cd stress than the other genotypes. The sensitive Tartary buckwheat genotypes also displayed greater leaf chlorosis and wilting, and, during longer hydroponic experiments, these genotypes became much more stressed than the others and withered earlier (Figure 1A). Pot experiments using Cd-contaminated soil confirmed that Cd stress could inhibit the growth of Tartary buckwheat in a similar way (Appendix A). These results indicate that the growth of Tartary buckwheat seedlings is affected by Cd stress in hydroponics and soil experiments and that some genotypes are more tolerant than others.

### 2.2. Effects of Cd Treatment on the Root System of Tartary Buckwheat Seedlings

The root systems were scanned to analyze the effects of Cd on root growth in more detail. Under the control conditions, the root growth of a more Cd-tolerant genotype, B3, and a more sensitive genotype, B6, was similar (Figure 2). Following Cd treatment, however, the total root length of the B6 genotype decreased to 63% of that of the Cd-free controls, and the number of root tips was only 49% of that of the controls. Genotype B3 was less affected by Cd treatment since its total root length was almost the same at 92% of the root length of the Cd-free control, and the number of root tips was 74% of that of controls. The reduction in root tips reflected the greater inhibition of lateral root growth in B6 compared with B3. The root volume, root surface area, and shape of the main roots of B6 and B3 were also affected by Cd, but the changes were significantly greater in B6 than in B3 (Appendix A). The roots of B6 appeared more rigid and twisted following Cd treatment than the roots of B3. These results indicate that Cd stress caused the main roots to become shorter and deformed and inhibited the emergence of lateral roots.

### 2.3. Cd Accumulation in Tartary Buckwheat

To investigate the accumulation and distribution of Cd in Tartary buckwheat seedlings, the levels of Cd in different tissues were determined by ICP-MS. In the absence of Cd, all nine genotypes contained low concentrations of Cd with ~15 μg/g DW in the roots and ~3 μg/g DW in shoots (Figure 3). The Cd concentrations increased after a five-day exposure to Cd in all the genotypes, but the final Cd concentrations in the roots of the sensitive genotypes, B6, B7, B8, and B9, (445 to 480 μg/g DW) were 18–27% greater than those in the more tolerant genotype, B3, (377 μg/g DW) (Figure 3A). The final Cd concentrations in the shoots of B1 and B3 after treatment were among the lowest at ~100 μg/g DW, while the concentrations in the shoots of B6 and B9 were almost 40% greater at ~136 μg/g DW (Figure 3B). These results show that that the uptake and accumulation of Cd in the root and shoot tissues of the sensitive genotypes of Tartary buckwheat was greater than that in the more tolerant genotypes.

### 2.4. Effect of Cd on Mineral Accumulation by Tartary Buckwheat

We evaluated how Cd stress affected the absorption of mineral nutrients. The concentration of the major nutrients under the control conditions varied among the nine Tartary buckwheat genotypes, and Cd treatment affected some of these. The largest changes occurred in Mn concentrations, which decreased by 23 to 65% in the roots and shoots, respectively, of all genotypes. The Cu concentrations were also affected by Cd and showed two-fold increases in the roots of all genotypes. More-variable trends were detected in other elements. For instance, Cd treatment tended to decrease K concentrations in the roots and shoots, reduce Mg concentrations in the roots, and increase Na concentrations in the roots (Figure 4 and Figure 5). However, the magnitude of these changes differed between genotypes depending on whether they had been scored as more resistant or more sensitive to Cd. For example, the average decrease in K and Mg concentrations in the roots of the three resistant genotypes (B1, B2, B3) was 8% and 7%, respectively, which were less than the decreases of between 18% and 24% measured in the sensitive genotypes (B6, B7, B8, B9). Similarly, the Ca concentrations in the roots of the three resistant genotypes showed significant increases after Cd treatment, whereas the Ca concentrations in the roots of all the sensitive genotypes decreased significantly. Finally, the Zn concentration in the roots of the three resistant genotypes was unaffected by Cd treatment, whereas it was significantly reduced in the sensitive genotypes B6, B7, B8, and B9. These results show that Cd can affect the uptake and accumulation of nutrients in the roots and shoots of Tartary buckwheat and that the more resistant genotypes are affected less than the sensitive genotypes.

### 2.5. Effects of Cd Stress on Cellular Ultrastructure of Tartary Buckwheat Seedlings

Cd stress not only affected the growth habit of plants but also altered the cell ultrastructure. These changes were compared in the Cd-sensitive B6 and Cd-resistant B3 genotypes. Under normal conditions, the cell structure of the root tip cells had typical structures with clear cell walls, nucleus, mitochondria (cristae process evenly distributed), vacuoles and other organelles, and well-shaped plasma membranes (Figure 6A,B). Following Cd treatment, the cell wall thickened and became separated from the plasma membrane; the nucleolus disintegrated; the mitochondria swelled; the inner chamber dilated; the number of the vacuoles increased; and black precipitates (osmiophilic granules) gradually appeared. Damage to the cell wall and changes to the overall cell morphology were greater in the sensitive genotype, B6, compared with those in the resistant genotype, B3, after the same Cd stress (Figure 6C–F). 

Under the control conditions, the chloroplast structure in the leaves was elliptic or fusiform in shape with well-displayed thylakoid membranes organized in the grana and stromal membranes. The structures of the plasma membrane, mitochondria, and nucleus were intact and well ordered (Figure 7A,B). Following Cd treatment, the subcellular structure of the leaf cells suffered damage. Similar to the case in the root cells, Cd stress caused plasmolysis and nucleolus disintegration. The chloroplast became more spherical, and the internal grana lamellar structure was disrupted with some grana thylakoids becoming loosely arranged in waves and exhibiting more thylakoid-free space. Osmiophilic granules were also found in the chloroplasts under Cd stress, and starch grains were observed in the chloroplasts of B6 but not of B3 (Figure 7C–F). These results indicate that Cd stress affected the growth and development of Tartary buckwheat and caused damage to the cell morphology and structure.

### 2.6. Analysis of Oxidative Stress Responses under Cd Stress

Oxidative damage is an early response to Cd toxicity in plants, and this is mainly reflected in the excessive production and accumulation of ROS and subsequent interactions such as peroxidation of membrane lipids. Therefore, we compared the production of superoxide anion radicals (O_2_^•−^) and accumulation of malondialdehyde (MDA) in the Cd-sensitive genotype, B6, with that in the more Cd-resistant genotype, B3. MDA reflects the antioxidant capacity and the degree of ROS production in plants and is used as an indirect indicator of oxidative damage. Under the control conditions, B6 and B3 contained similar concentrations of O_2_^•−^ of ~6 nmol/g/min FW in the root and leaf tissues (Figure 8). Following exposure to 30 µM of Cd for five days, the O_2_^•−^ concentration increased in the roots of both genotypes, but the increase in B6 was two-fold greater than that in B3 (Figure 8A). Under the control conditions, the MDA concentrations in the roots of B3 and B6 were similar (~1.6 μmol/g FW), and Cd treatment caused similar increases in both genotypes. In leaves, by contrast, the background MDA concentrations under the control conditions were almost three-fold greater in B3 compared with those in B6. Cd treatment increased the MDA levels in the leaves of B6 by 1.8-fold, but no significant changes occurred in the MDA levels in the leaves of B3 (Figure 8B). These results indicate that Cd toxicity induced oxidative stress in Tartary buckwheat seedlings and that the subsequent damage was greater in the Cd-sensitive genotype than in the Cd-resistant genotype.

Plant cells protect themselves from oxidative stress through the activity of antioxidant enzymes like SOD, POD, CAT, and APX, which scavenge the ROS to minimize cellular damage. The activities of these enzymes were measured in the roots and leaves of B3 and B6 plants with and without Cd treatment. Cd stress increased the activity of all the enzymes except APX (Figure 9). The largest differences between the two genotypes occurred in the Cd-induced changes to CAT activity, which showed a 3.6-fold increase in the roots of B3 compared with only a two-fold increase in those of B6 (Figure 9C). The Cd-induced changes to SOD and POD activities in the roots and leaves were also approximately two-fold greater in B3 than in B6 (Figure 9A,B). The finding that Cd stress induces the activity of several antioxidant enzymes to a greater extent in the resistant genotype than in the sensitive genotype suggests that these enzymes have a role in increasing the resistance of Tartary buckwheat to Cd stress.

## 3. Discussion 

Cd pollution is one of the most widespread environmental problems and has negative effects on many aspects of plant growth [32]. Previous studies have demonstrated that Cd stress induced abnormalities in many plant species [33,34,35]. In this study, the root growth and biomass accumulation of Tartary buckwheat seedlings were significantly reduced after exposure to Cd (Figure 1), which was consistent with results in rice [36] and alfalfa [37]. The inhibitory effect of Cd on growth might be attributed to the inhibition of cell division [38], disruption of chloroplast structure, and reduction of photosynthetic pigments, but other lesions are possible [6,39]. Furthermore, short- and long-term exposure to Cd stress induces stomatal closing, reduces chlorophyll content, and inhibits photosynthesis in many plants [10,19]. Our electron microscope scanning results confirmed that Cd stress disrupted the ultrastructure of the root cells and chloroplast structure in leaves (Figure 6 and Figure 7). Damage to the chloroplasts would affect chlorophyll biosynthesis and lead to chlorosis, wilting, and even death in extreme cases [6,32]. 

Previous reports have found that Cd can inhibit root elongation, lateral root development, and root surface area in many plant species, including rice [16,18], wheat [19], corn, and soybean [40]. Our results support these findings on primary root growth and lateral root emergence (Figure 2A,C) in Tartary buckwheat. The lateral root emergence depends on the specific distribution of auxin concentrations along the primary root [41], so Cd could be interfering with auxin function directly or via the expression and activity of auxin transporters [17,42].

Cd stress injures plants in other ways. The shorter, thicker roots associated with Cd treatment are associated with damage to the root apices, which could directly disrupt cell division [43]. By more closely examining the cell structure, we demonstrated that Cd caused the thickening of the root cell walls, altered the organelle structure, and induced the separation of the plasma membrane from the cell wall (Figure 6). These results suggest that Cd stress affects the cellular ultrastructure and cell division, which alters the overall root architecture.

An important outcome of this study was the detection of genotypic variation for Cd tolerance among different sources of Tartary buckwheat. These differences were associated with differential plant growth (Figure 1 and Figure 2), Cd accumulation (Figure 3), mineral nutrition (Figure 4 and Figure 5), and oxidative stress management (Figure 8 and Figure 9) in the tolerant and sensitive genotypes. It is unclear what controls the differences in Cd uptake, but others have suggested that it involves the secretion of organic compounds from the roots [44] or modification to the boundary cells near the root tips [45]. Root exudates can influence the pH and microbial communities in the rhizosphere, which could chelate Cd and minimize its harmful interactions with the root tissues [46]. The lower accumulation of Cd in tolerant genotypes might relate to reduced protein-facilitated transport of Cd across membranes; reduced damage occurring to the root tissues, which allows less Cd to enter; or the maintenance of a higher growth rate, which dilutes the Cd concentration in tissues. The Cd concentrations were much greater in the roots than in the shoots, and this has been attributed to apoplastic barriers and the binding of Cd in the cell walls of the roots, which impedes its radial movement to the stele [47]. Cd could also be sequestered in the vacuoles of the root cells, which also limits its transport to the shoots [48]. These mechanisms along with the possible release of organic chelatants need to be investigated further to determine whether they can explain any of the genotypic differences in Cd tolerance reported here. 

The effects of Cd stress on nutrient absorption vary among plant species. For instance, Cd toxicity significantly reduced the concentration of K and Ca, Mg, Zn, Fe, Mn, and B in barley roots [49] and reduced N, Ca, Mg, and P contents in both the roots and shoots of alfalfa [13]. The present study found significant changes in the final mineral concentrations as well and demonstrated that the magnitude of these changes was different in Cd-tolerant and -sensitive genotypes of Tartary buckwheat (Figure 4 and Figure 5). The concentration of Ca in Cd-tolerant plants increased after exposure to Cd stress, while it decreased in Cd-sensitive materials (Figure 4C,G). Tolerant genotypes maintain Mg, K, Zn, and Mn concentrations in their roots better than the sensitive genotypes do under Cd stress (Figure 4 and Figure 5). A previous study suggested that the competitive relationship between Cd and Ca absorption was caused by their net charges and ionic radii [6]. Interestingly, the lateral roots play an important role in the absorption of water and the uptake and storage of nutrients [50,51]. Therefore, the much greater inhibition of lateral root growth in sensitive genotypes, compared with that in tolerant genotypes, might partly explain the differences in final nutrient concentrations. 

Several studies have shown that Cd-induced oxidative stress led to the oxidation of lipids and major macromolecules in cells and interfered with antioxidant defense systems [21,23,52,53,54]. Consistent with the research of Lu et al. [30], Cd stress increased the content of MDA and ROS including H_2_O_2_, and O_2_^•−^ in Tartary buckwheat. Moreover, we found that the increase in MDA and O_2_^•−^ concentrations was lower in B3 (more tolerant) than in B6 (more sensitive) in both the root and leaf when compared to the controls (Figure 8). This reflects a lower level of oxidative injury in the more Cd-tolerant plants. Antioxidant enzymes are important for scavenging excessive ROS production to prevent damage to cells. Therefore, the finding that the tolerant genotypes showed higher SOD and POD activities in the roots and leaves than the sensitive genotypes and displayed higher CAT activities in the roots following Cd treatment (Figure 9) could indicate that the activity of these enzymes is central to the mechanism of Cd tolerance in Tartary buckwheat. This is supported by some other studies that showed that silencing the *HvVPE3* gene in barley (which encodes a Cd transporter) increased the activities of antioxidant enzymes and enhanced Cd tolerance [54]. Future studies should investigate these mechanisms in buckwheat.

## 4. Material and Methods

### 4.1. Plant Materials, Growth Condition, and Treatment

A total of nine registered varieties and landraces of Tartary buckwheat were examined in this study (Table 1). Sixty seeds were pre-germinated by first placing them on damp tissues in Petri dishes overnight at 4 °C and then for 3 days at 28 °C. The germinated seeds (45 plants in each variety) were transferred to plastic containers with 2.4 L 1/2-strength Hoagland nutrient solution (pH 5.80). Soft plastic foam was used to secure the seedlings in holes made in the lids of the container. The containers were placed in a phytotron with the following conditions: 14 h/25 °C and 10 h/22 °C day–night cycle, 70% relative humidity, and ~300 μM m^−2^ s^−1^ light intensity. Each bucket was aerated with one pipe attached to aquarium air pumps (Appendix A). The solution’s pH was checked daily and adjusted as required using 1 M HCl or 1 M NaOH. All the solutions were replaced after three days, and after one week, some containers received control solution and others received 30 μM of CdCl_2_. After a further five days of growth, the seedlings were harvested for subsequent measurements. 

### 4.2. Measurement of Phenotypic Index

Plants of each line were removed from the bucket, laid out on a dark cloth, and photographed. The root systems were photographed with a root scanner (EPSON 10000 XL, Nagano, Japan), and WinRHIZOProLA2400 software (Regent, Québec City, QC, Canada) was used to collect data on the length of the main root, total root length, total number of root tips number, root area, and root volume. The shoots and roots were then separated and dried at 105 °C for 30 min and then for a further two days at 80 °C to obtain the dry weights (DWs). A single replicate of root length, root DW, and shoot DW was calculated from the average of four plants. 

### 4.3. Determination of Physiological Index

The oxidative stress induced by Cd stress was detected by the concentration of malondialdehyde via the thiobarbituric acid method [30], and the rate of super oxygen free radical (O_2_^−^) production using the hydroxylamine method [55]. The activity levels of superoxide dismutase (SOD), peroxidase (POD), catalase (CAT), and ascorbate peroxidase (APX) were determined as previously described [30]. 

### 4.4. Electron Microscopic Scanning on Cell Ultrastructure

The scanning electron microscopic images were collected using methods described previously [56]. Briefly, the roots were placed in pre-cooled 25 mmol/L EDTA-Na_2_ for 20 min to remove ions attached to the root surface. They were then rinsed with deionized water, and the surface water was removed with filter paper. The roots and the leaves (without the midrib) were cut into small pieces and placed in a bottle with 2.5% glutaraldehyde buffer solution. The air was then sucked out of the bottle with a syringe so that the samples were fully soaked in the buffer solution. Samples were fixed at 4 °C for 24 h. The fixed samples were rinsed in phosphate buffer (pH 7.2) for 15 min and post-fixed in 1% OsO_4_ at room temperature for 2 h before being rinsed several times with 0.1 mol/L phosphate buffer (pH 7.0). The fixed samples were dehydrated in an ascending series of ethanol concentrations (30%, 50%, 70%, 80%, 90%, and 95%, 10 min each) and 100% ethanol (three times, 7 min each). Finally, the samples were dehydrated in 100% acetone for 20 min and then treated with a 0.5:1 and 1:1 propylene oxide and ethanol mixture, and 100% propylene oxide in turn. The samples were then embedded and infiltrated with epoxy resin at 70 °C for 48 h.

Sections of the embedded samples were obtained with an LKB-V ultramicrotome (LKB Ultrascan XL, Bromma, Sweden) and double stained with uranyl acetate–lead citrate before being examined with a transmission electron microscope (HITACHI H7650, Tokyo, Japan) operating at 80 kV.

### 4.5. Determination of Cd and Other Elements in Tissues

The harvested shoots and roots were washed three times with ice-cold water and then dried under 80 °C. The contents of Cd and other minerals, including K, Na, Ca, Mg, Fe, Zn, Mn, and Cu, were measured by inductively coupled plasma mass spectrometry (ICP-MS, NexlON, 2000) (PerkinElmer, Waltham, MA, USA).

### 4.6. Statistical Analysis and Reproducibility

All comparisons included at least three independent replicates. The data were analyzed with analysis of variance tests followed by Duncan’s multiple range test in SPSS 21.0 software (IBM, Chicago, IL, USA). The figures were plotted using Origin software version 8.0 (OriginLab Corporation, Northampton, MA, USA). 

## 5. Conclusions

The present study demonstrated that Cd stress affects the plant growth, root development, nutrient absorption, Cd transport and distribution, ultrastructure, and antioxidant metabolism in Tartary buckwheat. The variation in Cd tolerance among the Tartary buckwheat genotypes was linked with the activity of antioxidant enzymes. Other mechanisms may also be involved and should be investigated in the future. These include possible roles for root exudates to prevent Cd from entering the plants. All candidate mechanisms need to be confirmed in segregating populations generated by crossing resistant and sensitive genotypes to help link the growth phenotypes with the physiology.

## Figures and Tables

**Figure 1 plants-13-01650-f001:**
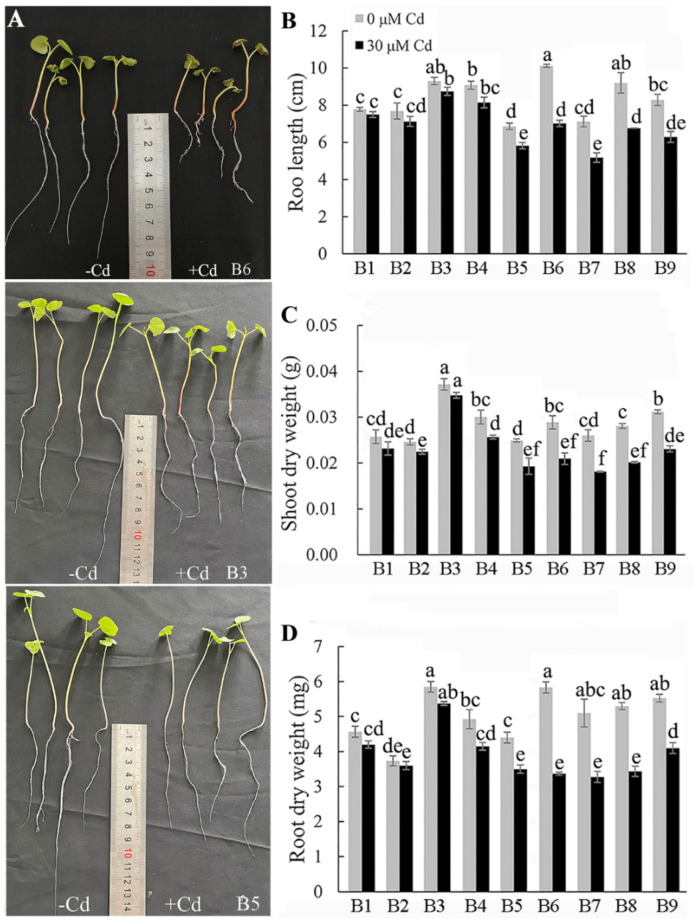
Effects of Cd stress on the growth of Tartary buckwheat seedlings. The effects of Cd stress on the (**A**) growth, (**B**) root length, (**C**) root dry weight, and (**D**) shoot dry weight. B1–B12 are different Tartary buckwheat genotypes. Values represent mean ± SD (*n* ≥ 12). Different letters indicate significant differences (*p* < 0.05) (Tukey’s test).

**Figure 2 plants-13-01650-f002:**
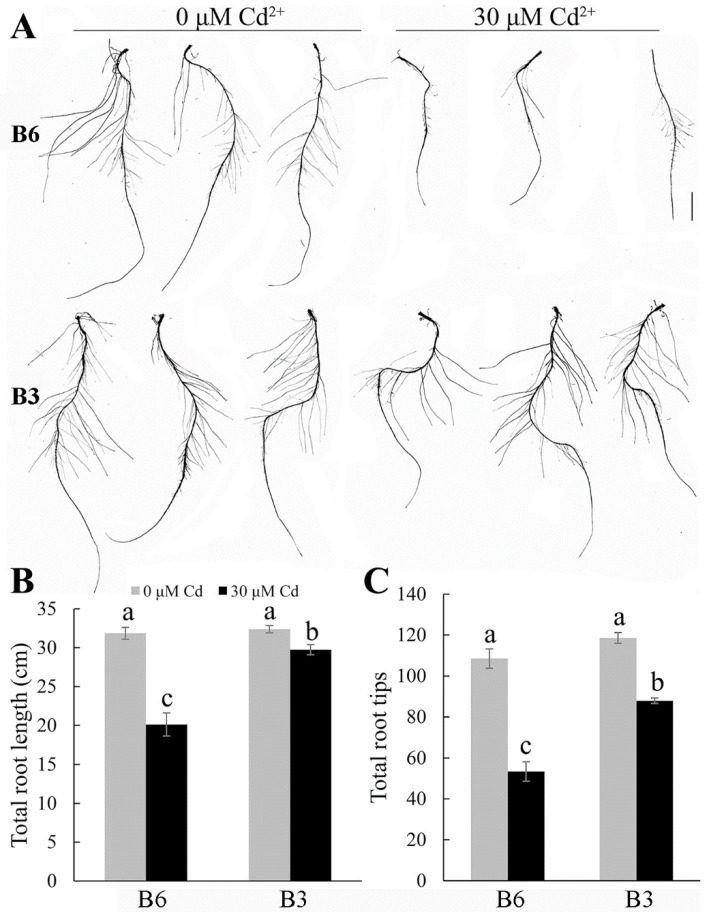
Effects of Cd stress on the root growth in different genotypes of Tartary buckwheat. (**A**) Root scanning, (**B**) total root length, and (**C**) total root tips of Tartary buckwheat seedlings treated with or without 30 μM CdCl_2_. Genotype B6 is relatively sensitive to Cd stress, and genotype B3 is relatively more tolerant of Cd stress. Scale bar: 1.0 cm. Values represent mean ± SD (*n* = 15). Different letters indicate significant differences (*p* < 0.05) (Tukey’s test).

**Figure 3 plants-13-01650-f003:**
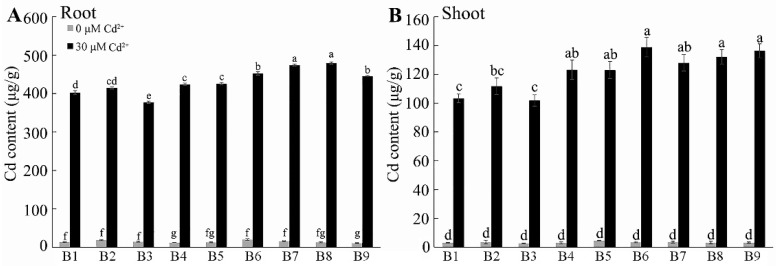
Cd accumulation in Tartary buckwheat under normal and Cd conditions. One-week-old seedlings were exposed to control (1/2 Hoagland solution, pH 5.80) and Cd (1/2 Hoagland solution with 30 μM CdCl_2_, pH 5.80) treatment for five days; then, both (**A**) root and (**B**) shoot Cd contents were measured. B1–B12 are different Tartary buckwheat genotypes. Different letters indicate the significant differences between different Tartary buckwheat resources at *p* < 0.05 based on Tukey’s test. Data are shown as mean ± SD (*n* = 15).

**Figure 4 plants-13-01650-f004:**
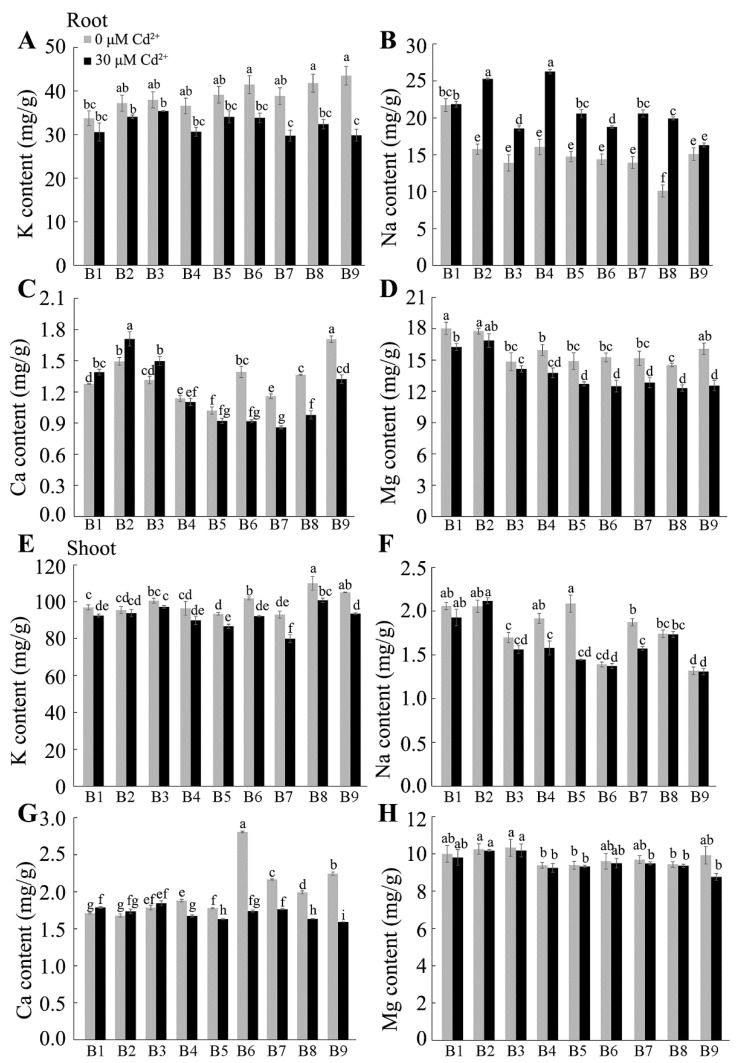
Effect of cadmium stress on the absorption of macroelements. The content of macroelements, including K, Na, Ca, and Mg, in both (**A**–**D**) roots and (**E**–**H**) shoots was analyzed under the control and Cd treatment. B1–B12 are different Tartary buckwheat genotypes. Different letters indicate the significant differences between different Tartary buckwheat resources at *p* < 0.05 based on Tukey’s test. Data are shown as mean ± SD (*n* = 15).

**Figure 5 plants-13-01650-f005:**
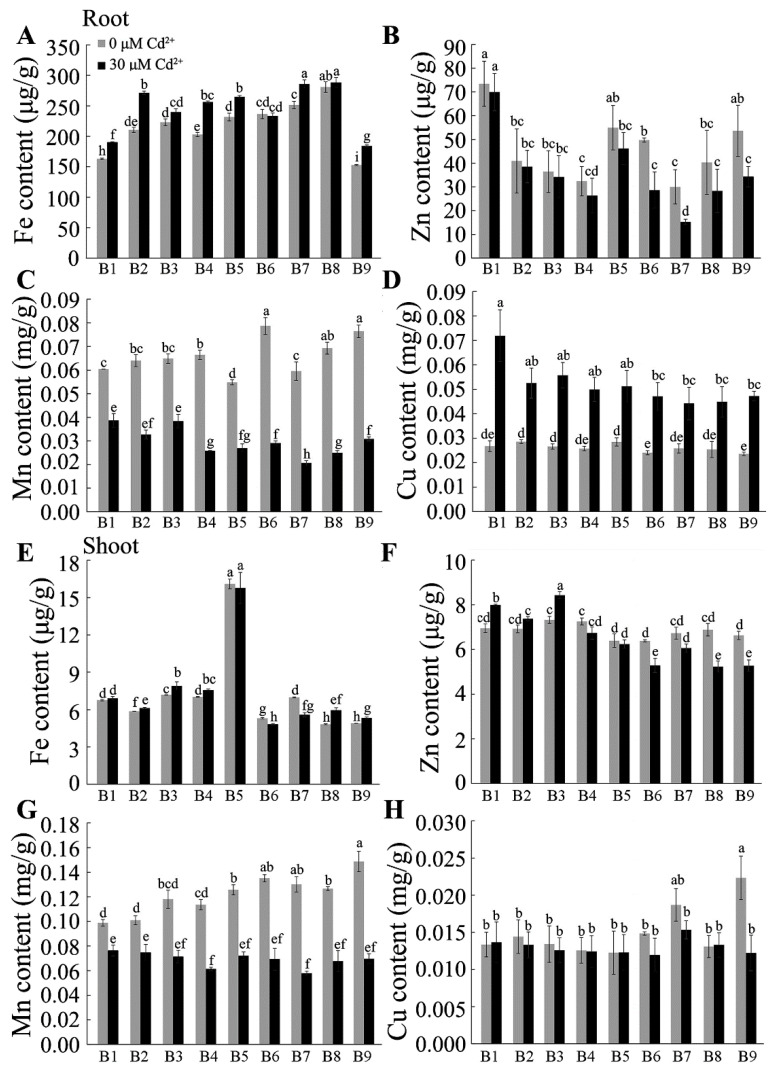
Plant microelement concentrations under the control and Cd-stress conditions. One-week-old seedlings were exposed to control (1/2 Hoagland solution, pH 5.80) and Cd (1/2 Hoagland solution with 30 μM CdCl_2_, pH 5.80) treatment for five days. The contents of Fe, Zn, Mn, and Cu in (**A**–**D**) roots and (**E**–**H**) shoots were determined. B1–B12 are different Tartary buckwheat genotypes. Values represent mean ± SD (*n* = 15). Different letters indicate the significant difference within each group at *p* < 0.05 according to Tukey’s test.

**Figure 6 plants-13-01650-f006:**
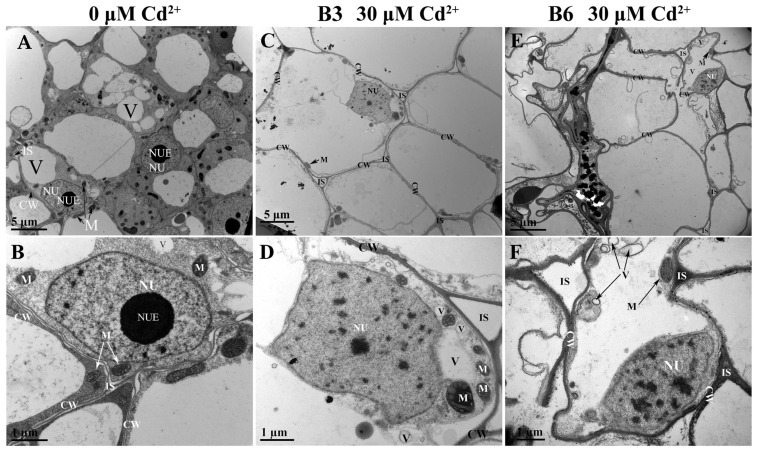
Ultrastructural changes to the roots of contrasting Tartary buckwheat genotype. Seedlings were treated with or without 30 μM CdCl_2_ for five days; then, the ultrastructure of Tartary buckwheat root under (**A**,**B**) control and (**C**–**F**) Cd stress was analyzed. B6 is a Cd-sensitive genotype, and B3 is a Cd-tolerant genotype of Tartary buckwheat. CW: cell wall; NUE: nucleolus; NU: nucleus; V: vacuole; M: mitochondrion; IS: intercellular space. Scale bar: 5 μm and 1 μm.

**Figure 7 plants-13-01650-f007:**
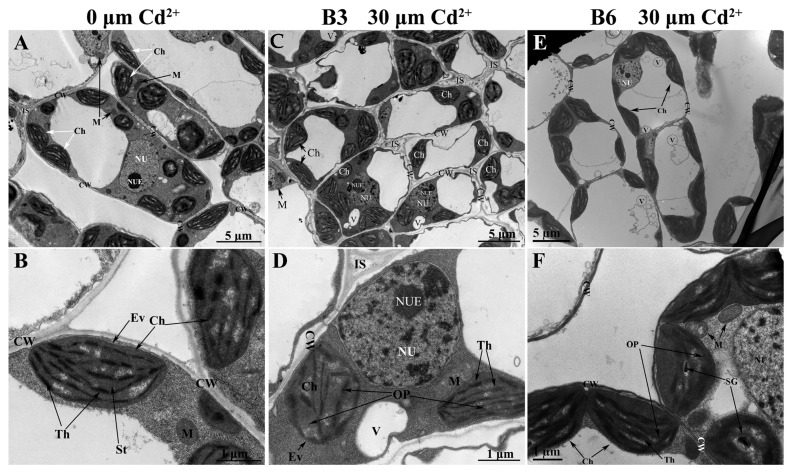
Variation in the leaf cell structure of Tartary buckwheat caused by Cd stress. The ultrastructure of Tartary buckwheat leaf under (**A**,**B**) control and (**C**–**F**) Cd stress was analyzed. Genotype B6 is relatively sensitive to Cd stress, and genotype B3 is relatively more tolerant of Cd stress. CW: cell wall; NUE: nucleolus; NU: nucleus; V: vacuole; M: mitochondrion; IS: intercellular space; Ch: chloroplast; OP: osmiophilic granule; SG: starch granule; Th: thylakoid. Scale bar: 5 μm and 1 μm.

**Figure 8 plants-13-01650-f008:**
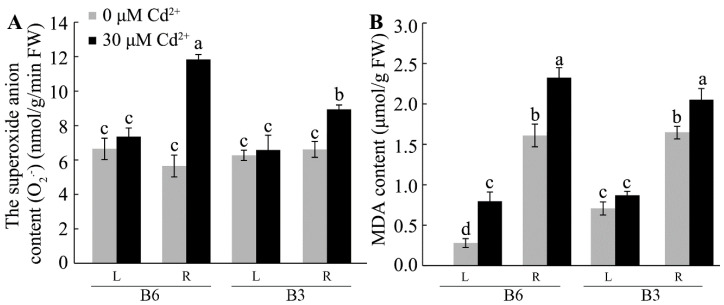
Oxidative stress of Tartary buckwheat seedlings induced by Cd stress. Roots and leaves of B3 (Cd-tolerant genotype) and B6 (Cd-sensitive genotype) buckwheat seedlings were collected after different treatments and evaluated for (**A**) superoxide anion (O_2_^•−^) and (**B**) malondialdehyde (MDA) contents. L represents leaf, and R represents root. Values represent mean ± SD (*n* = 15). Different letters indicate the significant difference within each group at *p* < 0.05 according to Tukey’s test.

**Figure 9 plants-13-01650-f009:**
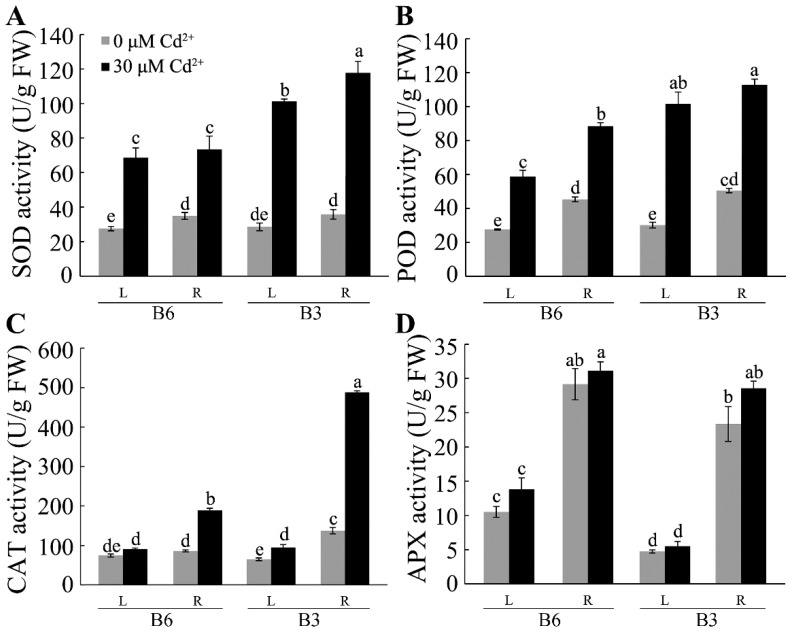
Activity of antioxidant enzymes in different genotypes of Tartary buckwheat exposed to Cd stress. Roots and leaves of B3 (Cd-tolerant genotype) and B6 (Cd-sensitive genotype) buckwheat seedlings were collected after different treatments and evaluated for the total activity of (**A**) SOD, (**B**) POD, (**C**) CAT, and (**D**) APX. L represents leaf, and R represents root. Values are means ± SD (*n* = 15). Different letters indicate significant differences at *p* < 0.05 (Tukey’s test).

**Table 1 plants-13-01650-t001:** Information on the Tartary buckwheat resources used and their relative ranking for Cd tolerance as determined from this study.

ID	Name	Origin	Genotype *
B1	Yanyuan 1	Baiwu Town, Yanyuan County, Sichuan, China	T
B2	Yanyuan 2	Benbangying Village, Yanyuan County, Sichuan, China	T
B3	Chuanqiao No. 1	Zhaojue Institute of Agricultural Sciences	T
B4	Qianku No. 6	Weining County Agricultural Science Research Institute	M
B5	Youku No.1	Chongqing Agricultural School, Southwest University, and Youyang County Agricultural Technology Extension Station	M
B6	Xiqiao No. 8	Xichang University	S
B7	Diku No. 1	Diqing Institute of Agricultural Sciences	S
B8	Jinqiao No. 6	Crop Research Institute of High and Cold Regions, Shanxi Academy of Agricultural Sciences; Datong Seed Management Station	S
B9	Yunqiao No. 2	Yunnan Academy of Agricultural Sciences	S

* The letters S, T, and M represent the Cd-sensitive, Cd-tolerant, and intermediate genotypes, respectively.

## Data Availability

Data are available on request to the corresponding author’s email with appropriate justification.

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
