# Peer review of "Effects of Cadmium Stress on Tartary Buckwheat Seedlings"

_plants, 2024, doi:10.3390/plants13121650_

Round 1

Reviewer 1 Report

Comments and Suggestions for Authors

Keywords: oxidative…?

Introduction:

Indicate mechanisms alleviating Cd toxicity in plants. Emphasize the ubiquitous role of antioxidant enzymes in mitigating different abiotic stresses (heavy metals, pesticides, drought, etc.). Refer to https://doi.org/10.1016/j.chemosphere.2022.136284

Results:

‘these genotypes were more likely to die’ – rephrase

‘Following Cd treatment however, total root length and the number of root tips of B6 were 63% and 49% of the controls, while those of B3 were 92% and 74% of controls, respectively’ – indicate precisely what the Authors mean?

Indicate that B3 was tolerant and B6 sensitive in the Figure 2

were determined by ICP-MS’ – remove from the footnote to Fig. 3

Increase the dimensions of Fig. 4 and 5 for better reading. Currently these Figures are illegible

‘of different Tartary buckwheat’ – rephrase the caption to Fig. 6

‘Seedling roots and shoots of B3 and B6 Tartary buckwheat were collected after exposure to 30 μM CdCl2 for five days and used to evaluate’ – remove from the footnote to Fig. 8

‘One-week-old B6 and B3 seedlings were treated with 30 μM CdCl2 for five days and’ - remove from the footnote to Fig. 9

Discussion:

Adjust the references to the Journal format

Explain why starch was formed in B6 sensitive genotype, but not in B3 tolerant

Page 12: B6 sensitive instead less tolerant

Page 13: function of the TaNHX2 gene

Materials and Methods:

How many seeds were sown to the containers?

How many plants per pot and per treatment?

Briefly describe the methods in 4.3

Indicate determined minerals in 4.5

Author Response

Responses to reviewers’ comments

#Reviewer 1

Keywords: oxidative…?

Reply: We have included “oxidative stress” in the Keywords.

Introduction:

Indicate mechanisms alleviating Cd toxicity in plants. Emphasize the ubiquitous role of antioxidant enzymes in mitigating different abiotic stresses (heavy metals, pesticides, drought, etc.). Refer to https://doi.org/10.1016/j.chemosphere.2022.136284

Reply: We thank the reviewer for this suggestion to mention general mechanisms in plants that alleviate Cd toxicity. We have now included the following text at L51-58 in the Introduction and added two references (references 25 and 26 in the revised manuscript):

“Certain species, and even genotypes within species, induce defensive mechanisms to protect them from Cd stress. These can be broadly divided into avoidance mechanisms and tolerance mechanisms. Avoidance mechanisms reduce Cd entry into the cytosol to maintain lower tissue concentrations by reducing transport processes and binding Cd in the cell wall. The tolerance mechanism better helps moderate the stress once the metal ions have entered the cytosol by chelating the heavy metals in less harmful complexes, compartmentalizing the ions to less sensitive organelles such as the vacuole or by activating antioxidative defense pathways (Baruah et al 2023). Antioxidant enzymes are involved with mitigating oxidative damage caused by a wide range of abiotic stresses apart from heavy metals including drought and mineral toxicities (Iwaniuk et al 2022).

References added:

[25] Baruah, N., Gogoi, N., Roy, S., Bora, P., Chetia, J., Zahra, N., Ali, N., Gogoi, P., Farooq, M. Phytotoxic responses and plant tolerance mechanisms to cadmium toxicity. J Soil Sci Plant Nutr 2023, 23, 4805-4826.

[26] Iwaniuk, P., Kaczyński, P., Pietkun, M., Łozowicka, B. Evaluation of titanium and silicon role in mitigation of fungicides toxicity in wheat expressed at the level of biochemical and antioxidant profile. Chemosphere 2022, 308, 136284.

Results:

‘these genotypes were more likely to die’ – rephrase

Reply: This sentence was modified to read: “The sensitive Tartary buckwheat genotypes also displayed greater leaf chlorosis and wilting and, during longer hydroponic experiments, these genotypes became more much stressed than the others and withered earlier (see L81-82 in the “manuscript R1_with trace track”.).

'Following Cd treatment however, total root length and the number of root tips of B6 were 63% and 49% of the controls, while those of B3 were 92% and 74% of controls, respectively’ – indicate precisely what the Authors mean?

Reply: Thank you – we agree this requires clarifying. The sentence now reads: “Following Cd treatment, however, the total root length of the B6 genotype decreased to 63% of the Cd-free controls and the number of root tips was only 49% of the controls. Genotype B3 was less affected by Cd treatment since its total root length was almost the same at 92% of the Cd-free control and the number of root tips was 74% of controls” (see L93-96 in the manuscript R1_with trace track).

Indicate that B3 was tolerant and B6 sensitive in the Figure 2

Reply: Thank you – agreed. We have made this clear in the figure legend of Fig. 2

'were determined by ICP-MS’ – remove from the footnote to Fig. 3

Reply: Thanks for your advice. We agreed this was unnecessary and removed those words from the footnote in Figure 3.

Increase the dimensions of Fig. 4 and 5 for better reading. Currently, these Figures are illegible

Reply: This is a good suggestion and changes have been made to make these figures easier to read. See responses to the Editor.

'of different Tartary buckwheat’ – rephrase the caption to Fig. 6

Reply: We have now modified this title to read. “Ultrastructural changes to the roots of contrasting Tartary buckwheat genotypes”

‘Seedling roots and shoots of B3 and B6 Tartary buckwheat were collected after exposure to 30 μM CdCl2 for five days and used to evaluate’ – remove from the footnote to Fig. 8. 'One-week-old B6 and B3 seedlings were treated with 30 μM CdCl2 for five days and' - remove from the footnote to Fig. 9

Reply: We agreed this was unnecessary and removed those words from legend. It was modified as suggested to read: “Roots and leaves of B3 and B6 buckwheat seedlings were collected after different treatments and evaluated for . . . . . . .”

Discussion: Adjust the references to the Journal format

Reply: Thanks a lot for your advice. We have adjusted the reference to the Journal format already.

Explain why starch was formed in B6 sensitive genotype, but not in B3 tolerant

Reply: According to the study of He et al (2005), the starch grains (SGs) in the chloroplast were a stress response. They speculate that the accumulation of SGs might be related to the obstruction of photosynthate transport caused by abiotic stress. Similarly, we speculated that the accumulation of SGs in B6 chloroplasts was also caused by this reason, because it was observed in the more Cd-sensitive genotype B6 but not the more Cd-tolerant genotype B3. In addition, the SGs in the chloroplast need to be converted into glucose before they can be transported to other parts of the plant (Smith et al., 2003). Cd stress might reduce the activity of enzymes related to this process, resulting in starch conversion blocked and accumulation in chloroplasts.

Reference

He T, Wu X.M., Zhang C.N., Wang X.R., Jia X.F. Characteristics of starch grains in chloroplast of five alpine plants. J Wuhan Bot Res (in Chinese) 2005, 23(6): 545-548.

Smith A.M., Zeeman S., Niittylä T., Kofler H., Smith S.M. Starch degradation in leaves. J Appl Glycosci 2003 50(2): 173-176.

Page 12: B6 sensitive instead less tolerant. Page 13: function of the TaNHX2 gene

Reply: We revised the description as your suggestion, please check it in the manuscript R1_with trace track (L289-292).

Reply: We apologize for this typographical error and thank the Reviewer for picking it up. The gene we were referring to here was actually HvVPE3 and not TaNHX2. The HvVPE3 protein localizes to endoplasmic reticulum and is involved in Cd transport. The reference [54] we refer to shows that by silencing the HvVPE3 gene in barley, the activities of antioxidant enzymes increased and Cd tolerance was enhanced. We have not rewritten this and corrected the text (from L289).

Materials and Methods: How many seeds were sown to the containers? How many plants per pot and per treatment?

Reply: 60 seeds were sown in the containers, and 45 plants per pot and treatment. We have added the corresponding description to the manuscript.

Briefly describe the methods in 4.3. Indicate determined minerals in 4.5

Reply: We revised the description as your suggestion, please check it in the manuscript R1_with trace track.

Reviewer 2 Report

Comments and Suggestions for Authors

The paper entitled "Effects of cadmium stress on Tartary buckwheat seedlings" shows the effect of cadmium on the growth of seedlings.  Of interest was the variation of genotypes and their response to Cadmium.  No evidence was given that variation in genotypes typically has this effect for metal uptake.  It would be beneficial if a literature search was completed to describe this.  Is this the first report of these differences by genotype?  This paper focused on seedlings, the physiological effects and the molecular mechanisms.  It was very well written, and all the acronyms were defined (albeit later in the paper). It would be interesting to see the results when all the genotype results were averaged together to compare with and without Cd exposure.  Would there be a significant difference then?  Also, is there any speculation of which chelates are involved with differences found for the genotypic differences.  Nine genotypes were investigated.  Was it known which ones would be sensitive, tolerant or intermediate before the experiment began?  How are these genotypes different besides coming from different locations?  The results are very well presented; the methods are very clearly stated. 

Comments on the Quality of English Language

Very good English. 

Author Response

Responses to Editors’ comments

#Reviewer 2

The paper entitled "Effects of cadmium stress on Tartary buckwheat seedlings" shows the effect of cadmium on the growth of seedlings.  Of interest was the variation of genotypes and their response to Cadmium. No evidence was given that variation in genotypes typically has this effect for metal uptake.  It would be beneficial if a literature search was completed to describe this.  Is this the first report of these differences by genotype?  This paper focused on seedlings, the physiological effects and the molecular mechanisms.  It was very well written, and all the acronyms were defined (albeit later in the paper). It would be interesting to see the results when all the genotype results were averaged together to compare with and without Cd exposure.  Would there be a significant difference then?  Also, is there any speculation of which chelates are involved with differences found for the genotypic differences.  Nine genotypes were investigated.  Was it known which ones would be sensitive, tolerant or intermediate before the experiment began?  How are these genotypes different besides coming from different locations?  The results are very well presented; the methods are very clearly stated.

Reply: We thank the Reviewer for these positive comments. Responses to their specific questions are shown below.

(1) No evidence was given that variation in genotypes typically has this effect for metal uptake.  It would be beneficial if a literature search was completed to describe this. Is this the first report of these differences by genotype?

Reply: Thanks for the question. Cd tolerance is a large field and a thorough literature review is beyond the scope of this study. However, to answer the specific question, yes, genotypic variation in Cd tolerance has been reported in other species including the grasses like maize (Wu et al 2023; https://doi.org/10.1002/jsfa.12303) and legumes like common bean (Bahmani et al 2020; https://doi.org/10.1016/j.ecoenv.2020.110178).

(2) Also, is there any speculation of which chelates are involved with differences found for the genotypic differences.

Reply: This is an interesting question that we did not focus on in the current study. Work has been published on this topic in some species. It appears that phytochelatins are important chelatants. For instance, Arabidopsis mutants that lack a phytochelatin (PC) synthesis system, have increased sensitivity to Cd. Also, overexpression of the wheat phytochelatin (PC) synthase gene in tobacco plants increases Cd resistance (see Chen et al 2019, Sci Rep 9, 86; https://doi.org/10.1038/s41598-018-36228-z).

(3) Was it known which ones would be sensitive, tolerant or intermediate before the experiment began? How are these genotypes different besides coming from different locations?

Reply: Thank you. This is an important question and needs clarifying. We did not know there was variation in Cd among these genotypes prior to this study. This is made clear in the legend in Table 1.

Round 2

Reviewer 1 Report

Comments and Suggestions for Authors

The Authors have corrected the manuscript. I have no more comments.

Author Response

Dear Reviewer:
Thank you so much for your helpful comments on our manuscript. We would like to express our sincere appreciation to the anonymous reviewers for their careful reading and invaluable comments on our manuscript. We have revised the manuscript in accordance with the editor's and the reviewers’ comments. It has been proof-read once again to minimize typographical, grammatical, and bibliographical errors.